

# Gene expression in *Tribolium castaneum* life stages: Identifying a species-specific target for pest control applications

Lindsey C. Perkin[1] and Brenda Oppert[2]

[1] Southern Plains Agricultural Research Center, USDA, Agricultural Research Service, College Station, TX, United States of America
[2] Center for Grain and Animal Health Research, USDA, Agricultural Research Service, Manhattan, KS, United States of America

## ABSTRACT

The red flour beetle, *Tribolium castaneum*, is a major agricultural pest of post-harvest products and stored grain. Control of *T. castaneum* in stored products and grain is primarily by fumigants and sprays, but insecticide resistance is a major problem, and new control strategies are needed. *T. castaneum* is a genetic model for coleopterans, and the reference genome can be used for discovery of candidate gene targets for molecular-based control, such as RNA interference. Gene targets need to be pest specific, and ideally, they are expressed at low levels for successful control. Therefore, we sequenced the transcriptome of four major life stages of *T. castaneum*, sorted data into groups based on high or low expression levels, and compared relative gene expression among all life stages. We narrowed our candidate gene list to a cuticle protein gene (CPG) for further analysis. We found that the CPG sequence was unique to *T. castaneum* and expressed only in the larval stage. RNA interference targeting CPG in newly-emerged larvae caused a significant ($p < 0.05$) decrease in CPG expression (1,491-fold) compared to control larvae and 64% mortality over 18 d. RNA-Seq of survivors after 18 d identified changes in the expression of other genes as well, including 52 long noncoding RNAs. Expression of three additional cuticle protein genes were increased and two chitinase genes were decreased in response to injection of CPG dsRNA. The data demonstrate that RNA-Seq can identify genes important for insect survival and thus may be used to develop novel biologically-based insect control products.

## INTRODUCTION

The red flour beetle, *Tribolium castaneum,* is a pest of stored grain commodities. Traditional control methods for *T. castaneum* and other stored product beetles are rapidly becoming less effective, mostly because insect populations are developing resistance to pesticide treatments (*Boyer, Zhang & Lemperiere, 2012*). For example, storage pests around the globe are developing high resistance levels to one of the most common grain fumigants, phosphine (*Opit et al., 2012*; *Pimentel et al., 2010*). Thus, there is a need for new pest control strategies, and we are evaluating genetic-based treatments with target specificity and less

Corresponding author
Lindsey C. Perkin,
lindsey.perkin@ars.usda.gov

damage to the environment, including the application of RNA interference (RNAi; *Baum et al., 2007*; *Noh, Beeman & Arakane, 2012*).

*T. castaneum* is a useful model to identify candidate genes because it has a sequenced genome (*Tribolium Tribolium Genome Sequencing Consortium, 2008*) and responds well to RNAi when dsRNA is injected (RNAi; *Brown et al., 1999*; *Tomoyasu & Denell, 2004*; *Aronstein, Oppert & Lorenzen, 2011*; *Miller et al., 2012*). Oral RNAi was successful in *T. castaneum* larvae against genes encoding vATPase (*Whyard, Singh & Wong, 2009*), inhibitor of apoptosis (*Cao, Gatehouse & Fitches, 2018*), and a voltage-gated sodium ion channel (*El Halim et al., 2016*). However, we and others have not had success with oral RNAi in *T. castaneum* (unpublished data, *Palli, 2014*). Many factors may influence RNAi efficacy in insects, such as target sequence specificity, concentration and length of dsRNA, persistence of silencing effect in the target pest, and nucleases counteracting the effect of dsRNA (*Huvenne & Smagghe, 2010*; *Allen & Walker, 2012*; *Lomate & Bonning, 2016*; *Guan et al., 2018*; *Cao, Gatehouse & Fitches, 2018*). In the meantime, we have focused on the identification of gene targets with low expression requiring lower doses of dsRNA, and those that are expressed in critical feeding stages (larvae and adults) to improve the efficacy of oral RNAi.

The iBeetle project (http://ibeetle-base.uni-goettingen.de) conducted a large-scale RNAi screen in *T. castaneum* larvae and pupae with injected dsRNA, and various phenotypes were observed, including mortality and developmental abnormalities (*Schmitt-Engel et al., 2015*). From this screen, eleven genes were identified as potential pest control targets (*Donitz et al., 2015*; *Ulrich et al., 2015*). These genes encode mostly products with GO terms related to the proteasome, and mortality was observed after injection of larvae, pupae and adults. Some of these genes have orthologs in other species, such as *Drosophila melanogaster, Aedes aegypti, Apis mellifera,* and *Acyrthosiphon pisum*, and therefore RNAi or other molecular strategies based on these genes may not be species-specific and may not be practical for insect control. *Knorr et al. (2018)* selected 50 genes from the iBeetle study that resulted in ≥ 90% mortality to evaluate for RNAi in *Diabrotica vigifera vigifera* and found that feeding adults dsRNA targeting 20 genes resulted in mortality, and 36 retarded growth. While this work is important to highlight genes that have application across species, the usefulness in the field may be limited due to off-target effects.

Previously, we applied RNA-Seq to the major developmental stages of *T. castaneum* to identify highly expressed cysteine peptidase genes in feeding stages (larvae and adults) with likely digestive functions (*Perkin, Elpidina & Oppert, 2016*). A similar approach was used across the life stages of *D. melanogaster* to reveal new transcribed regions and small non-coding RNAs, demonstrating that transcriptome sequencing can discover new elements and help to annotate genomes of even the most well studied insects (*Graveley et al., 2011*). Transcriptome sequencing of developmental stages of the non-model oriental fruit fly, *Bactocera dorsalis*, and the harlequin bug, *Murgantia histrionica,* led to a list of candidate developmental and insecticide resistance genes for downstream studies (*Shen et al., 2011*; *Sparks et al., 2017*).

In the present study, we examined gene expression patterns in *T. castaneum* life stages (adults, eggs, larvae, and pupae), and used statistical analysis to sort expression patterns into

groups of low and high expression. Our hypothesis was that genes with low expression can be more successfully targeted with RNAi or other applied control methods because the dose of dsRNA required to knockdown transcript expression will be lower, and highly expressed genes and gene expression patterns may be useful in understanding the physiology of the insect, or to identify promoters for transgenic applications. Previously, we found that some genes with high expression (i.e., those encoding cysteine peptidases) were important to insect physiology and were duplicated in the genome, and knockdown of these genes induced a compensation response and upregulation of other closely related genes (*Perkin, Elpidina & Oppert, 2016*). Through screening of potential targets in the present study, we identified a unique cuticle protein gene (CPG) expressed in larvae, and we provide functional evidence for its potential as a molecular target for pest control. We further evaluated the effect of CPG RNAi on the expression of other genes, defining new potential metabolic connections between CPG and other genes.

## MATERIALS & METHODS

### Insects

Insect strains and collection protocols were as previously described (*Perkin, Elpidina & Oppert, 2016*). Briefly, the *T. castaneum* lab colony used in this study was originally collected from a grain bin in Kansas and has been maintained on 95% wheat flour and 5% brewer's yeast at 28 °C, 75% R.H., 0L:24D. Insects were subcultured from the laboratory colony, and specific life stages were removed. Adults were collected at 3–7 days post-eclosion and were mixed sex. Eggs were sifted from diet 24–48 h after oviposition and were separated from the fine particulate with a brush. Larvae were collected at a late instar stage that was actively feeding (approximately 14 days post hatch). Pupae included those with pigmentation in the eye, but not the elytra.

### Library preparation and RNA sequencing of life stages

RNA was collected as three independent biological replicates from mixed sex groups of each life stage (10 adults, approximately 500 eggs, 10 larvae, and 10 pupae per replicate). Tissue was pulverized in TRIZOL (BulletBlender, Next Advance Inc., Averill Park, NY, USA) at speed 8 for 2 min with RNAse-free ziroconium oxide beads. RNA extraction and purification were with a Zymo mini prep kit (Irvine, CA, USA). From the total RNA, DIRECTbeads (Agilent, Santa Clara, CA, USA) were used to isolate polyA mRNA, and libraries were made with a 200 bp RNA-Seq v2 kit (Life Technologies, Grand Island, NY, USA). Samples were sequenced on 318v2 chips on the Ion Torrent Personal Genome Machine (PGM, Life Technologies). Each run provided approximately 1–5 million reads, with a total of 5–12 million reads per life stage (*Perkin, Elpidina & Oppert, 2016*). Life stage sequences were deposited at NCBI SRA as part of BioProject PRJNA299695.

### Data analysis

Relative gene expression was analyzed using ArrayStar (Lasergene Genomics Suite v14, DNASTAR, Madison, WI, USA) by mapping reads to the NCBI Tcas5.2 genome build. Read counts were normalized by Reads Per Kilobase of template per Million mapped reads

(RPKM, (*Mortazavi et al., 2008*)). All data were filtered to groups with low expression (RPKM greater than 2, less than 8 $\log_2$; i.e., the $\log_2$ of the RPKM value was between 2 and 8, not inclusive) and high expression (RPKM greater than 8 $\log_2$) for comparisons and biological screening. We use a cutoff of RPKM greater than 2 to avoid underrepresented genes that can skew data.

Gene ontology (GO) terms were obtained and enrichment of GO terms was analyzed in BLAST2GO PRO (version 4.1.5, Valencia, Spain, *Götz et al., 2008*). Over-representation tests were performed with GO terms associated with genes in low and high categories in each life stage, using the program g:Profiler and the g:GOSt function, which provides statistical enrichment analysis (http://biit.cs.ut.ee/gprofiler/index.cgi; (*Reimand, Arak & Vilo, 2011*)). In g:Profiler, results were limited to *T. castaneum* and the main GO categories: biological process (BP), molecular function (MF) and cellular component (CC). Significance was determined using the False Discovery Rate (FDR) multiple-testing correction threshold of 0.05 (*Benjamini & Hochberg, 1995*). Enriched GO terms also were obtained from the pairwise comparison of gene expression in larvae injected with cuticle protein gene (CPG, LOC103313766) dsRNA compared to mock-injected, where the test set was the significant ($p < 0.05$) differentially expressed genes, and the reference set was expression from all genes. GO terms were limited to $p < 0.05$ (after FDR correction) and collapsed to the most specific description of each term.

## Candidate gene selection

Genes were first filtered to only include those with expression levels between RPKM 2 and 8 ($\log_2$). The resulting gene list included 181 genes. We then filtered to only include genes expressed at low levels in larvae because a field application would rely on genes expressed during the most active feeding stage, reducing the list to 139 genes. Our next filter was a manually curated gene set and was based on GO analysis and gene annotations, selecting genes with a function potentially necessary for survival.

The final filter submitted potential genes to the program OfftargetFinder (*Good et al., 2016*). This program checks gene specificity by searching 101 arthropod transcriptomes, 21mers at a time, to the 1000 Insect Transcriptome Evolution (1KITE) dataset (http://www.1kite.org) and is specifically designed to identify potential problems in designing dsRNA for insect control. The algorithm provides visualization of specific regions of the target gene that may cause off-target effects and allows for dsRNA to be designed to regions with increased specificity to the target pest. This filter limited our gene list to only three genes. We selected CPG as our primary target because the 3′ region of CPG was unique to *T. castaneum*, and this region was used to design primers for dsRNA.

## RNAi

Primers for dsRNA targeting CPG were designed similar to *Perkin, Elpidina & Oppert (2017a)* via Primer-BLAST (http://www.ncbi.nlm.nih.gov/tools/primer-blast/) using default parameters, and primer specificity was evaluated against the *T. castaneum* genome. A primary set of primers was designed to amplify CPG, and secondary primers were designed to amplify the 3′ region to minimize off-target effects.

The primary set of primers (Forward: ATAATCAAGCCCGTTTCCAACA, Reverse: AATCACGACTACAAACATTCTTAGG) amplified CPG in 25 μl PCR reactions using genomic DNA template from the lab strain (2.5 μl 10X buffer, 2.0 μl 2.5 mM dNTP, 0.5 μl 10 μM forward primer, 0.5 μl 10 μM reverse primer, 0.125 μl JumpStart AccuTaq DNA polymerase (Sigma-Aldrich, Saint Louis, MO, USA), 2 μl genomic DNA, 17.375 μl nuclease-free water) and thermal cycle conditions as specified by the JumpStart AccuTaq DNA polymerase product information guide (denaturation for 30 s at 95 °C, 30 cycles of 95 °C for 30 s, primer annealing at $T_m$ 55.5 °C for 30 s, and extension at 68 °C for 30 s, final extension at 68 °C for 5 min, and hold at 4 °C). The product was assessed on a 1% agarose E-gel (Thermo Fisher, Waltham, MA, USA) to ensure the amplified region was the correct size.

A second round of PCR was done with the secondary set of primers (Forward: TATTCGTCTGTCGTCGCTCC, Reverse: AATCACGACTACAAACATTCTTAGG) for specific targeting of CPG. The secondary PCR reaction was 100 μl and the product from the primary PCR reaction was used as the template (10 μl 10× buffer, 8 μl 2.5 mM dNTP, 2 μl T7 forward primer, 2 μl T7 reverse primer, 0.5 μl JumpStart AccuTaq DNA polymerase, 10 μl primary PCR product, 67.5 μl nuclease-free water). The thermal reaction profile was as follows: denaturation for 30 s at 95 °C, 5 cycles of 95 °C for 30 s, primer $T_m$ for 30 s, 72 °C for 30 s, 29 cycles of 95 °C for 30 s, primer $T_m$ plus 5 °C for 30 s, and 72 °C for 30 s, extension for 5 min at 68 °C and hold at 4 °C. The secondary PCR primers had a T7 construct attached to the 5′ end (TAATACGACTCACTATAGGG), and thus 5 °C was added to the second round of amplification as suggested in the kit manual. The secondary PCR products were checked again via 1% agarose E-gel for appropriate length (150 bp) and sufficient amplification.

PCR products from the secondary amplification were used to make dsRNA via MEGAscript T7 kit (Thermo Fisher) according to the kit protocol (8 μl dNTP mix, 2 μl 10X reaction buffer, 8 μl secondary PCR products with T7 construct, 2 μl enzyme mix), and after a 6 h incubation period at 37 °C with constant mixing, products were purified via MEGAclear kit (ThermoFisher). Size and quantity were verified on a digital nanophotometer (Implen, Westlake Village, CA, USA) and TapeStation (Agilent).

## Micro-injected dsRNA

Immediately before injection, dsRNA solution was mixed with blue dye (1:20) to aid in visualization of the injected liquid as in *Perkin, Elpidina & Oppert (2017a)*. Larvae used in injections were briefly put on ice, transferred to a microscope slide with double-sided tape for stabilization, and the entire slide was placed on top of a small ice block. Thirty individuals were injected with 200 ng of dsRNA in triplicate. CPG-dsRNA was injected into larvae weighing approximately 1 mg. Each treatment injection had a corresponding control injection of dye only (Mock) and a non-injection control (Control). Injections were with a Drummond Nanoject (Drummond Scientific Co., Broomall, PA, USA) set at 69 nl and fast injecting speed with a "bee-stinger" needle. Needles were made with 3.5 Drummond glass capillary tubes (3-000-203-G) and a micropipette puller (Model P-97; Sutter Instrument Co., Novato, CA, USA) using the program: heat 750, pull 100, velocity

8, time 250, and pressure 500. After injection, each group was allowed to recover for 2 h at room temperature, and then were covered with diet (95% wheat flour, 5% brewer's yeast) and kept at 28 °C, 75% R.H., 0L:24D. All treatments and controls were followed for 18 days to monitor mortality and/or developmental abnormalities.

## Library preparation and RNA sequencing and data analysis after RNAi

We validated knockdown of CPG with RNA-Seq as previously (*Perkin, Gerken & Oppert, 2017b*). Eight individuals from each treatment and control were flash frozen in liquid nitrogen on day 18 post injection. Total RNA was extracted by Trizol (Thermo Fisher Scientific, Waltham, MA USA) and Quick-RNA Mini Prep kit (Zymo Research, Irvine, CA USA). Libraries were made on the NeoPrep (200 bp insert) and sequenced on a Mi-Seq (2 × 300, Illumina, San Diego, CA USA). Three independent biological replicates of all groups and 3–4 technical replicates of CPG dsRNA-injected larvae were sequenced (metrics are found in File S5). Sequence reads were deposited at NCBI SRA as PRJNA520884.

Arraystar (DNAStar, Madison, WI USA) was used to calculate RPKM and fold change between treatment and control groups by mapping reads to the Tcas5.2 genome. We also used Arraystar for statistical analysis, student t-tests between treatments and ANOVA among treatments. Genes were filtered to only include those that had >8-fold difference between treatments, and significant at $p < 0.01$ after FDR correction. To classify ncRNA, sequences were submitted to RNAcon (website: http://crdd.osdd.net/raghava/rnacon/submit.html) for prediction using SVM scores (default of 0.0; *Panwar, Arora & Raghava, 2014*).

## Cuticle protein tree

NCBI RefseqRNA was searched with term ''cuticle protein'' and limited to *T. castaneum*, which returned a list of 160 genes which we then manually curated to only include relevant genes with a chitin domain (pfam00379). This resulted in a total of 125 sequences in the final data set. Protein sequences from the selected genes were used as input in MEGA-× (version 10.0.5; *Kumar et al., 2018*). Protein sequences were aligned using ClustalW. The evolutionary history of cuticle proteins was inferred using the Maximum Likelihood method, JTT matrix-based model, and 500 bootstrap iterations (*Jones, Taylor & Thornton, 1992*), using the tree with the highest log likelihood.

# RESULTS

In the following sections, transcriptome datasets of gene expression in four developmental stages of *T. castaneum* were analyzed by sorting genes into categories of high and low expression levels. Gene Ontology (GO) terms were associated with each group, and overrepresented terms were identified by enrichment analysis. A cuticle protein gene was selected as a gene expressed at low levels only in larvae and was evaluated via RNAi as a gene with critical function for insect survival.

## Life stage expression analysis—genes expressed at high levels

Analysis of *T. castaneum* genes with high expression levels (RPKM > 8, $\log_2$) in all life stages indicated there were 101, 290, 238, and 54 genes uniquely expressed in adults, eggs, larvae,

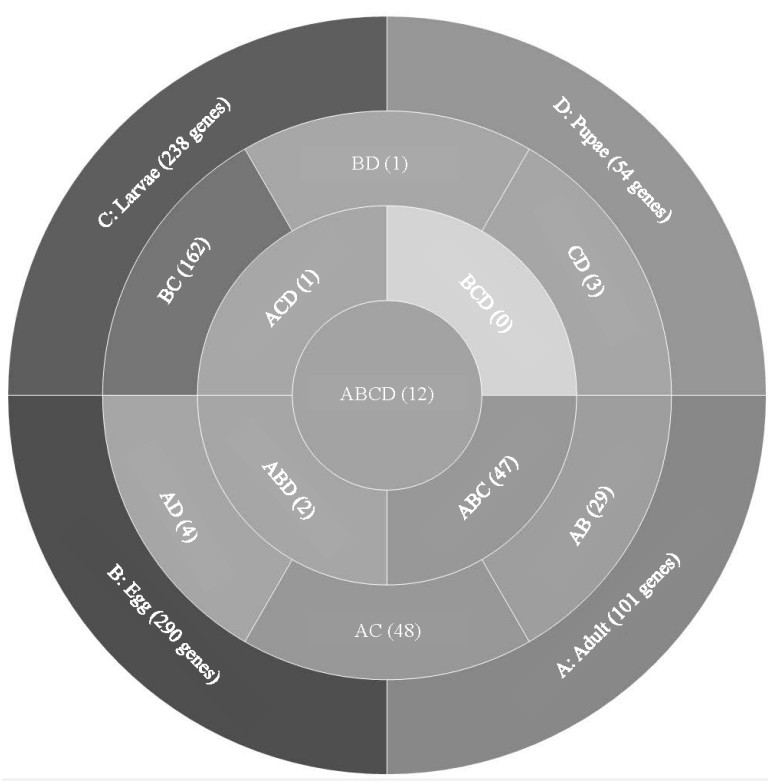

**Figure 1** **Venn diagram comparing genes expressed in *T. castaneum* life stages: adult, egg, larvae, and pupae with high expression values (>8 log$_2$ RPKM).** Shades of grey indicate the relative number of genes in each comparison where dark grey indicates a higher number of genes and lighter shades represent comparisons with fewer genes. The number in parenthesis is the number of genes in each comparison.

and pupae, respectively (Fig. 1, File S1). Eggs and larvae stages shared the most commonly expressed genes (162), followed by adults and larvae (48), and adults and eggs (29). Genes common to eggs and larvae included 44 ribosomal proteins (large and small subunits), indicative of the high degree of protein production occurring in these stages. Adults and larvae feeding stages shared two digestive cathepsin L genes (LOC659441 and LOC659502; (*Martynov et al., 2015*; *Perkin, Elpidina & Oppert, 2016*) and two chitinase precursor genes (*Cht8* and *Cht11*). Only LOC663906 encoding a single-stranded DNA-binding protein was commonly expressed in eggs and pupae. Twelve genes were highly expressed in all stages, including those encoding two elongation factors, three ribosomal proteins, a cathepsin L (26-29-p), glyceraldehyde-3-phosphase dehydrogenase 2, NADH dehydrogenase subunit 2, ATP synthase, cytochrome c and cytochrome b, and I-rRNA (Table 1).

In the highly-expressed gene group, the top Molecular Function (MF) GO terms for each life stage varied, but adults and larvae had shared term "nucleotide binding", whereas eggs and larvae shared "hydrolase activity" (Fig. 2). "Oxidoreductase activity" was common to larvae, pupae and adults. An enrichment analysis found 16 MF terms belonging to "structural constituent of the ribosome" ($p = 1.07 \times 10^{-11}$) and 62 with "catalytic activity" ($p = 0.028$) enriched only in adults.

**Table 1  Genes that were expressed at high (top) and low (bottom) levels in all life stages of *T. castaneum*.**

| Name | Description | Average RPKM | | | |
|---|---|---|---|---|---|
| | | **Adult** | **Egg** | **Larvae** | **Pupae** |
| LOC656235 | Elongation factor 1-gamma | 1,560 | 1,334 | 657 | 6,278 |
| RpS3 | Ribosomal protein S3 | 3,953 | 2,663 | 1,475 | 263 |
| 26-29-p | Cathepsin L | 369 | 281 | 261 | 5,710 |
| LOC659992 | 40S ribosomal protein S29 | 1,135 | 1,443 | 626 | 3,238 |
| LOC663023 | Glyceraldehyde-3-phosphate dehydrogenase 2 | 2,405 | 812 | 2,135 | 2,375 |
| LOC660435 | 60S ribosomal protein L11 | 2,673 | 2,134 | 1,096 | 6,999 |
| Efa1 | Elongation factor 1-alpha | 2,253 | 2,541 | 1,762 | 3,576 |
| ND2 | NADH dehydrogenase subunit 2 | 1,081 | 1,457 | 1,000 | 401 |
| ATP6 | ATP synthase F0 subunit 6 | 5,442 | 4,960 | 4,555 | 270 |
| COX3 | Cytochrome c oxidase subunit III | 4,267 | 3,923 | 6,559 | 857 |
| CYTB | Cytochrome b | 3,181 | 4,829 | 7,123 | 558 |
| I-rRNA | I-rRNA | 16,635 | 11,824 | 34,216 | 11,020 |
| LOC103315067 | Uncharacterized | 5.91 | 57.1 | 7.30 | 18.7 |
| LOC103312214 | Uncharacterized | 10.3 | 89.0 | 8.11 | 24.6 |
| LOC103315070 | Uncharacterized | 17.5 | 94.1 | 16.9 | 26.9 |
| LOC107399196 | Uncharacterized | 25.5 | 181 | 9.69 | 22.7 |
| LOC103312419 | Maternal protein tudor | 5.25 | 15.3 | 5.34 | 5.45 |
| LOC103312455 | poly(A) polymerase type 3 | 9.11 | 65.5 | 13.0 | 15.0 |
| LOC663288 | THO complex subunit 2 | 7.55 | 61.8 | 8.69 | 10.6 |
| LOC659780 | Transmembrane protein 35 | 240 | 7.72 | 241 | 179 |
| LOC103313244 | Sentrin-specific protease | 10.5 | 74.6 | 35.0 | 9.45 |

We also examined the most highly expressed genes in each developmental stage. The top 10 most highly expressed genes in adults included those encoding an odorant binding protein, protamine, and cytochrome c oxidase subunit II (RPKM 7,694–20,300), and the highest expressed genes in eggs had similar function, including chemosensory protein 2, I-rRNA and cytochrome c oxidase subunit I and II (RPKM 7,960–26,611; File S1). These genes also were highly expressed in larvae, along with cathepsin L genes involved in digestion of cereal proteins (RPKM 13,248–43,202; (*Martynov et al., 2015*; *Perkin, Elpidina & Oppert, 2016*)). Highly expressed genes in pupae encoded two cuticular proteins and a conserved cathepsin L, 26-29-p (RPKM 3,576–13,818).

## Life stage expression analysis—genes expressed at low levels

A comparison of genes with low expression levels (RPKM between 2–8 $\log_2$) revealed expression of 273 genes unique to adults, 167 to eggs, 139 to larvae, and 14 to pupae (Fig. 3; File S2). Adults and larvae shared 138 genes including chitin deacetylase 7 (*Cda7*) and two chitinase precursors (*Cht4* and *Cht9*). Larvae and pupae shared a single gene, LOC664054 encoding pathogenesis-related protein 5, which also was expressed in adults but at higher levels and thus was not part of the low expression group. Nine genes expressed at low levels were common across all life stages, four of which had uncharacterized functions (Table 1).
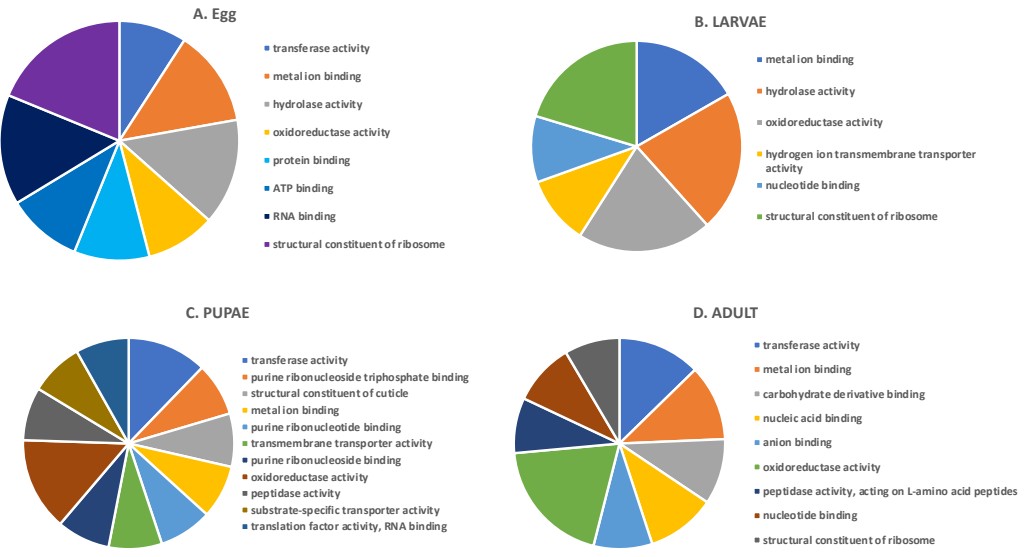

**Figure 2** **Top GO terms for genes with high expression in each life stage.** Each pie chart identifies the top GO terms for each life stage; (A) Egg, (B) Larvae, (C) Pupae, (D) Adult.

The other five included genes encoding maternal protein tudor (*Tctud*), poly(A) polymerase type 3, THO complex subunit 2, transmembrane protein 35, and sentrin-specific protease.

A GO analysis of each group of unique genes indicated differences in top level functions among stages, with more similar expression in the adults and larval stages, sharing MF terms "serine-type endopeptidase activity", "oxidoreductase activity", and "transition metal ion binding" (Fig. 4). However, an enrichment analysis indicated that serine-type endopeptidases were overrepresented only in the adults stage ($p = 0.005$), and substrate-specific transporter activity was overrepresented in the larval stage ($p = 0.04$). Still, these two life stages shared 138 genes expressed at low levels, not unexpected since both stages are active and feeding. Pupae had the least number of genes and GO terms and included those with MF term "serine-type endopeptidase activity", and two unique terms: "NADH dehydrogenase activity", and "structural constituent of cuticle". Enrichment analysis for pupae indicated overrepresentation of negative regulation of programmed cell death (BP; $p = 0.002$), which was due to reads mapping to the gene LOC663274 that encodes fas apoptotic inhibitory molecule 1, consistent with cell division and tissue rearrangement occurring during this stage. Genes expressed in eggs were associated with terms "sequence-specific DNA binding" and "transcription factor activity sequence-specific DNA binding". The most enriched categories in eggs were BP anatomical structure development ($p = 0.002$), sex determination ($p = 0.011$), and cell differentiation ($p = 0.006$), consistent with highly complex developmental processes during embryogenesis.

## Selection of candidate gene

To find candidate genes for RNAi, we manually curated genes unique to each life stage and expressed at low levels (File S2). Six cuticle genes were expressed only in larvae, but unfortunately many cuticle genes have orthologs in other species. However, a BLAST of
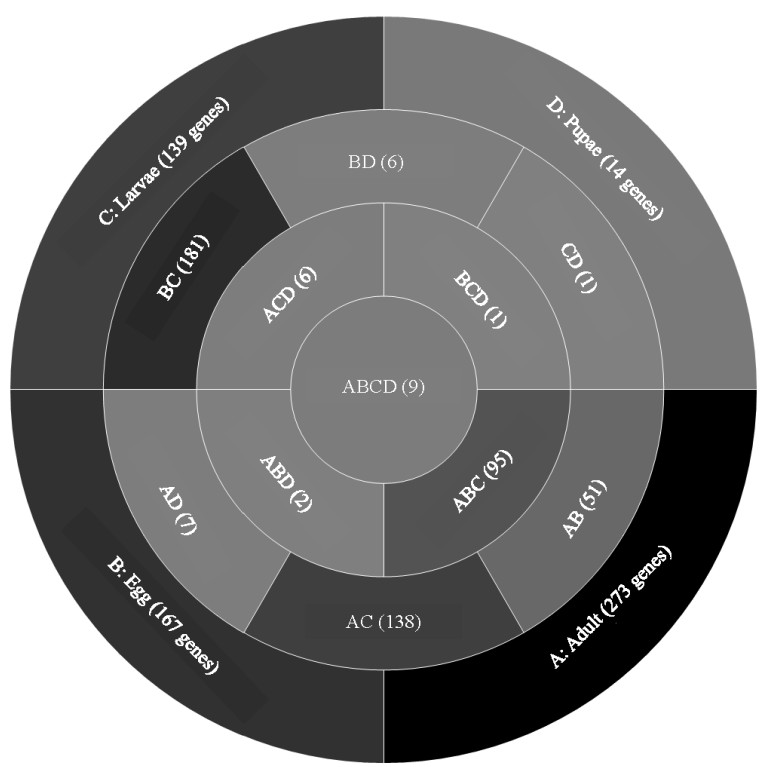

**Figure 3** **Venn diagram comparing genes between all *T. castaneum* life stages: adult, egg, larvae, and pupae with low expression values (2–8 log$_2$ RPKM).** Shades of grey indicate the relative number of genes in each comparison. Dark grey indicates a higher number of genes and lighter shades represent comparisons with fewer genes. The number in parenthesis is the number of genes in each comparison.

CPG (LOC103313766) indicated no hits to other species (data not shown). In examining previous data (*Morris et al., 2009*), CPG was found to be moderately expressed in the larval gut (relative to other gut-expressed genes). Unfortunately, this gene was not analyzed by the iBeetle project (*Schmitt-Engel et al., 2015*). We then submitted the sequence to OffTargetFinder and found hits to only four other insect species which were focused to the 5′ and middle regions, leaving the 3′ as a unique portion of the sequence (Fig. S1). So, we chose LOC103313766 CPG to evaluate as a potential candidate, since it met our criteria of uniquely expressed in *T. castaneum* larvae, contained regions where dsRNA would minimize off target effects, and was expressed in the larval gut.

### RNAi validation of CPG

To demonstrate that this was a biological target, we used RNAi to knockdown expression of CPG in *T. castaneum* larvae. We injected dsRNA into early stage larvae (2nd–3rd instar), which we hypothesized would have the most impact on transcript abundance. Knockdown and off target effects were evaluated by another round of RNA-Seq on RNA isolated from the injected larvae and controls (mock-injected "Mock" and noninjected "Control") 18 d post injection. The RNAi treatment significantly reduced the expression of CPG target transcript 1,491-fold compared to Control ($p = 2.17 \times 10^{-7}$) and 284-fold compared to

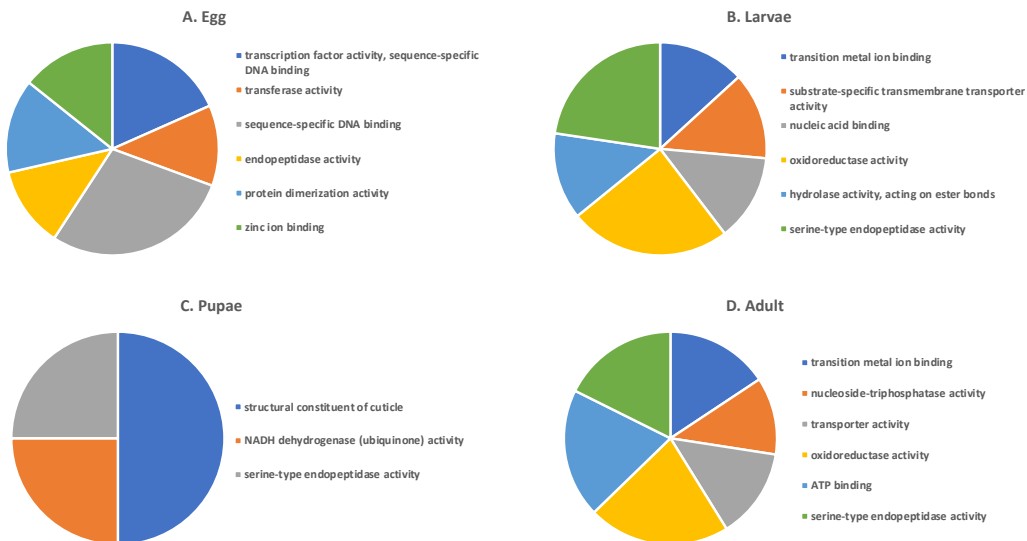

**Figure 4** **Top GO terms for genes with low expression in each *T. castaneum* life stage.** Each pie chart identifies the top GO terms for each life stage; (A) Egg, (B) Larvae, (C) Pupae, (D) Adult.

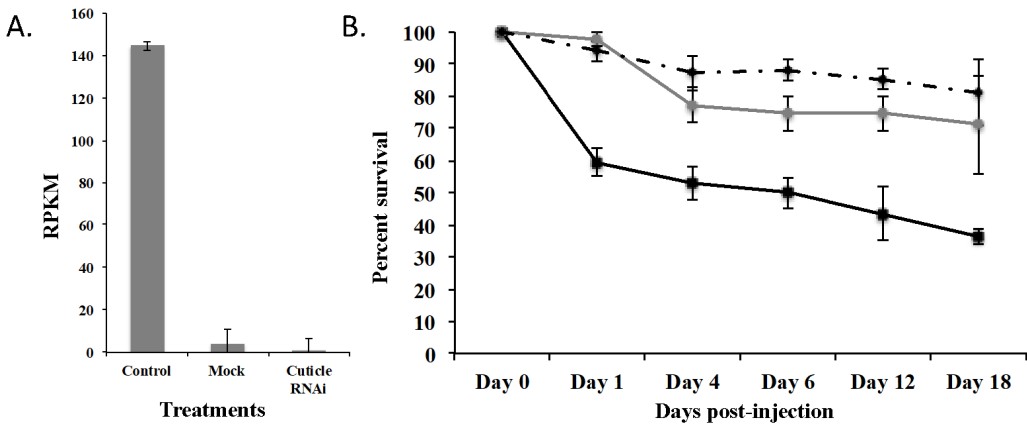

**Figure 5** **Effects of injection of CPG dsRNA in *T. castaneum* larvae.** (A) Average expression of CPG in control, mock, and dsRNA-injected *T. castaneum* larvae 18 d following RNAi application. (B) Percent survival of *T. castaneum* larvae following RNAi treatment.

Mock ($p = 7.14 \times 10^{-7}$). An ANOVA comparing the RPKM values of Mock, Control, and CPG treatments also was significantly different (Fig. 5A; $p = 2.64 \times 10^{-5}$). The large decrease in expression was biologically significant because 64% mortality was observed in the CPG dsRNA treatment group compared to the non-injected control at day 18 post-injection (Fig. 5B; $p = 0.057$). We also observed 29% mortality in Mock, but it was not significant when compared to Control mortality (19%; $p = 0.79$).

RNA-Seq analysis of larvae injected with CPG dsRNA compared to Mock resulted in 449 significantly differentially expressed genes ($p < 0.01$ and at least 8-fold change;

Supplemental Fig. 2 and File S3). Of those genes, 100 were uncharacterized, and 20 were annotated as hypothetical proteins. Most (62%) had decreased expression, and the 10 most highly repressed genes were uncharacterized except the target gene and LOC100142553 adenylate kinase. Of those with increased expression, the most highly increased was *Jtb*, which encodes an orphan receptor with unknown function, LOC662244 F-box SPRY that contributes to ubiquitin-protein transferase activity, and *BBIP1* which is a chromosome associated protein. Other up regulated genes related to transcription/translation included LOC107398339 (splicing factor for mRNA), LOC103313738 (a repressor of developmentally regulated gene expression), and LOC664045, involved in chromatin silencing. In response to knockdown of CPG, three other cuticle protein genes were significantly increased: LOC100141875 cuticle 19-like, LOC103313752 cuticle protein 70, and LOC655183 cuticle protein.

There were 52 genes in this dataset that were annotated as noncoding RNA (ncRNA), 25 with increased and 27 with decreased expression (File S4). All were preliminarily characterized as long ncRNAs since they were greater than 200 nt. All ncRNAs were screened through RNAcon (*Panwar, Arora & Raghava, 2014*) to predict the classification of each. There were 10 SSU-rRNA5, three Intron-GP-1, and one IRESe; the remaining sequences were below the default Support Vector Machines (SVM) threshold (0.0) and were classified as coding mRNA. However, these sequences were characterized by NCBI as ncRNA, probably due to unusual structure. Regardless, our transcriptome data suggests that ncRNA are actively transcribed in response to RNAi, as has been described previously (*Ji et al., 2015*).

A GO term enrichment analysis of significantly decreased gene expression in response to CPG RNAi identified functions involved in chitin breakdown (BP GO:0006032 and MF GO:0004568; $p = 0.04$ and 0.04, respectively). LOC107398196 and *Cht13*, both encoding chitinases, were responsible for these enriched GO terms. In contrast, the GO term aspartic-type endopeptidase activity ($p = 8.28 \times 10^{-5}$) was significantly enriched in up regulated genes.

## Characterization of CPG

To gain insight into the function of the CPG target gene, a maximum likelihood tree was constructed with *T. castaneum* protein sequences annotated as 'cuticle protein' (Fig. 6). CPG predicted protein (XP_008196095, highlighted with a yellow box) was most similar to cuticle proteins 16.5 (XP_015833062) and 19.8 (XP_976285). Interestingly, the target CPG protein also was found in the same major clade as the up regulated cuticle protein 70 gene LOC103313752 (XP_008196069, highlighted in red), and found in close proximity on LG3 (Table 2). Another cuticle protein gene with increased expression, LOC655183 (XP_008190752) also on LG3, was related to those encoding resilin proteins. The protein product of the cuticle gene with highest increased expression, LOC100141875 (XP_001809825), was related to cuticle protein 7 (XP_008193006) encoded by LOC656347, and both genes were found on LG5.

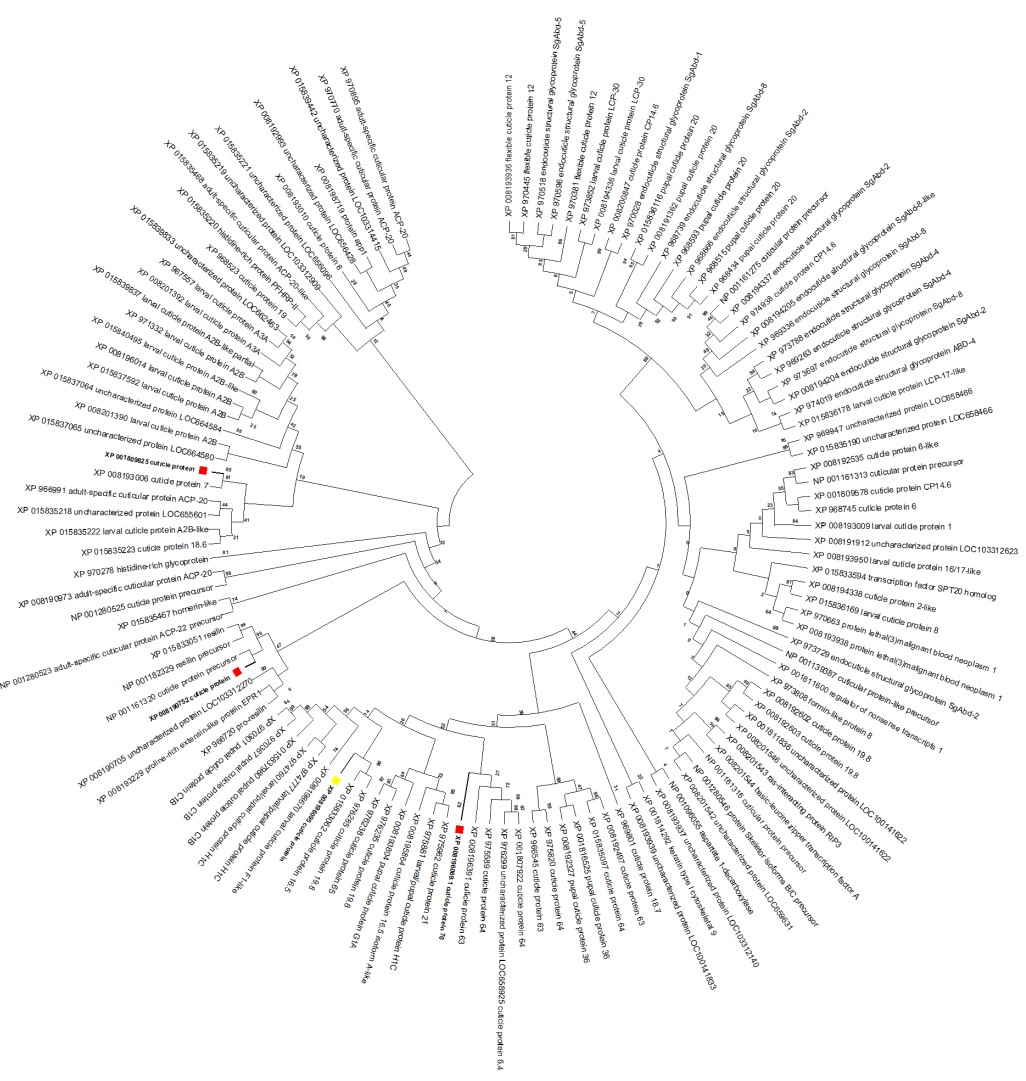

**Figure 6** ***T. castaneum.*** **cuticle protein maximum likelihood tree.** CPG (XP_008196095) is highlighted with a yellow box, and proteins from the three cuticle genes with increased expression (XP_008196069, XP_001809825, XP_008190752) are highlighted with a red box.

# DISCUSSION

Current control methods for *T. castaneum* and other stored product pests are becoming less effective due to development of insecticide resistance (*Collins, 1990*; *Jagadeesan et al., 2012*; *Opit et al., 2012*). Alternative management strategies are limited, and new molecular-based controls are needed because they offer increased efficacy and selectivity, and targeted approaches to combatting resistance. There are major hurdles in developing molecular-based strategies in *T. castaneum*. The first step is to identify a target gene that can be effectively knocked-down. The target gene should be specific to the pest or a narrow range of pests. The most difficult challenge is delivery, whether it is topical or oral RNAi, or

**Table 2  Differentially expressed cuticle genes in *T. castaneum* larvae injected with CPG dsRNA.** Shaded row indicates target gene.

| Gene | LG | Coordinate Start | Coordinate Stop | Protein | Annotation | Fold Change | Expression |
|------|----|------------------|-----------------|---------|------------|-------------|------------|
| LOC103313766 | 3 | 4099231 | 4115167 | XP_008196095 | Cuticle protein | 1,491 | ↓ |
| LOC658110 | 3 | 4097672 | 4112513 | XP_015833062 | Cuticle protein 16.5 | 2.51 | ↓ |
| LOC658489 | 3 | 4077823 | 4109423 | XP_976285 | Cuticle protein 19.8 | 1.44 | ↓ |
| LOC103313752 | 3 | 4114536 | 4126952 | XP_008196069 | Cuticle protein 70 | 10.8 | ↑ |
| LOC655183 | 3 | 2446367 | 2620651 | XP_008190752 | Cuticle protein | 4.01 | ↑ |
| LOC100141875 | 5 | 3608978 | 3616499 | XP_001809825 | Cuticle protein | 16.9 | ↑ |
| LOC656347 | 5 | 3612511 | 3613210 | XP_008193006 | Cuticle protein 7 | 4.89 | ↑ |
| LOC107399215 | 2 | 6796409 | 6962942 | XP_015840495 | Larval cuticle protein A2B-like | 25.1 | ↓ |

another mechanism. In this study, we evaluated molecular targets by injection of dsRNA prior to attempts of topical or oral delivery.

An RNA-Seq analysis of the four major life stages of *T. castaneum* identified twelve genes that were highly expressed in all developmental stages. One gene was a cathepsin L named 26-29-p. Cathepsin L genes in *T. castaneum* are part of a gene expansion group that are expressed at high levels in adults and larvae for the digestion of cereal proteins (*Goptar et al., 2012*; *Oppert et al., 2010*; *Oppert et al., 2003*; *Oppert et al., 1993*; *Perkin, Elpidina & Oppert, 2016*; *Perkin, Elpidina & Oppert, 2017a*; *Vinokurov et al., 2009*). Based on phylogenetic analysis, 26-29-p is conserved across insects and may be the ancestral cathepsin L gene, but it most likely functions in development and immunity (*Perkin, Elpidina & Oppert, 2016*). While this gene may not be useful for species-specific pest control because of the conservation of sequence among insects, its function in *T. castaneum* and other insects is of biological interest, as it appears to provide critical function(s) throughout development.

Nine genes were commonly expressed at low levels in all developmental stages, including *Tctud* (maternal protein tudor). *Tctud* has been found in many eukaryotes and has been implicated in protein-protein interactions, where methylated substrates bind to the tudor domain. In *D. melanogaster,* tudor proteins are found in chromatin, small nuclear RNA assembly, RNA-induced silencing complex, and germ granules (*Ying & Chen, 2012*). Similar to *Tctud*, temporal expression data shows low to moderately high expression of *Dmtud* in early fly embryos through 30-day old adults in both males and females (flybase.org, modENCODE temporal expression data, (*Gramates et al., 2017*). Additionally, mutants of various alleles caused lethality of early embryos through maternal effects, and both male and female sterility (*Thomson & Lasko, 2004*); flybase.org, Summary of Phenotypes, (*Gramates et al., 2017*). We submitted *Tctud* nucleotide sequence to OffTargetFinder and found multiple 21mer hits to 35 other species (data not shown), and therefore the gene was not selected for RNAi.

We have found that validation of RNAi knockdown through RNA-Seq can provide unique insights into function. Previously, we used RNA-Seq to validate knockdown of aspartate 1-decarboxylase (*ADC*) in *T. castaneum* larvae, resulting in the identification of additional gene interconnectivity (*Perkin, Gerken & Oppert, 2017b*). Not only was the significant knockdown of *ADC* confirmed through RNA-Seq, a previously unknown

interaction between *ADC* and dopamine receptor 2 also was discovered. This information led to biological assays that determined that the reduction of *ADC* transcripts via RNAi resulted in adults with decreased mobility. Therefore, we used the same approach in the present study to evaluate knock down of a target gene and also understand the overall impact to the transcriptome of reduced gene expression.

To isolate a candidate gene that could be used as a molecular-based pest control target, we sought genes unique to a feeding stage (i.e., larvae or adults) and expressed at low levels. Through manual curation of life stage transcriptome data, we found CPG was uniquely expressed in larvae at low levels and lacked an ortholog in other species at the 3′ end of the gene; in fact, BLAST did not return any hits at the mRNA level. We were encouraged that this target also has potential as an oral RNAi product, since the gene was moderately expressed in the larval gut. The protein product of this gene, XP_008196095, is a peptide of 61 amino acids, with 41% identity to a neuropeptide-like precursor from the flesh fly, *Sacrophaga crassipalpis* (*Li, Rinehart & Denlinger, 2009*). Additionally, *Bhatia & Bhattacharya (2018)* found that knockdown of a related cuticle protein gene in the green peach aphid (*Myzus persicae*) through oral delivery of dsRNA expressed in *Arabidopsis thaliana* resulted in reduced fecundity. We did not investigate whether the *T. castaneum* target cuticle protein also had neuropeptide properties or whether reduced gene expression caused a reduction in fecundity, but our functional test with RNAi demonstrated that decreased gene expression in early stage larvae resulted in significant mortality compared to Control and Mock. Thus, this gene is a candidate target for oral delivery of dsRNA.

RNA-Seq analysis of the transcriptome after RNAi knockdown of CPG identified three other cuticle genes with increased expression and two chitinase genes with decreased expression compared to Mock. The cuticle genes with increased expression included one encoding cuticle protein 70 (XP_008196069), clustering in the same major clade as the target cuticle protein gene, suggesting similar function and perhaps an indication of redundancy in function as has previously been noted with gene expansion groups. In fact, this redundancy may explain survivors in the dsRNA-injected larvae, albeit a low percentage (32%), and it may complicate oral RNAi if there is compensation response similar to that previously identified in attempting to target cysteine protease genes via RNAi (*Perkin, Elpidina & Oppert, 2017a*). Another cuticle protein gene LOC655183 also was up regulated after RNAi treatment, and this gene product (XP_008190752) clustered with resilin proteins. Resilin is a specialized cuticle protein that is found in soft parts of the adult insect allowing for movement of wings (*Andersen & Weis-Fogh, 1964*). LOC655183 is an ortholog to a gene in *Drosophila melanogaster* encoding cuticular protein 56F, which is expressed in the imaginal discs of late larvae (flybase.org, (*Gramates et al., 2017*). The protein product of another cuticle gene with increased expression, LOC100141875 (XP_001809825), clustered with cuticle protein 7 (XP_008193006), an ortholog to one found in the carapace cuticle of juvenile horseshoe crab, *Limulus Polyphemus* with a high tyrosine content (*Ditzel, Andersen & Højrup, 2003*).

Opposite in response to the increased expression of cuticle protein genes, *Cht13,* a chitinase, was decreased in response to the injection of CPG dsRNA. *Cht13* is predicted to be part of group IV chitinases expressed in the gut or fat body and expressed in response to

feeding in larval and adult stages (*Zhu et al., 2008*). All differentially expressed chitin-related genes are apparently interconnected through a regulatory pathway that responds to the loss of function of CPG by RNAi.

We were surprised at the number of long ncRNAs (52) that were differentially expressed at a significant level ($p < 0.01$) in surviving *T. castaneum* larvae that were injected with CPG dsRNA. Most ncRNA were not classified as typical ncRNAs and may represent new classes, or alternatively may be miss-annotated. However, differentially expressed long ncRNA have been implicated in human disease states, particularly cancer (*Ma et al., 2015*). Of relevance to this study, differentially-expressed long ncRNAs can impose epigenetic changes that alter the transcription of other genes, including silencing (*Tufarelli et al., 2003*). Therefore, we can speculate that the large number of differentially expressed genes in this study (449) may have been regulated in part by ncRNAs. While not to the same degree, ncRNAs were implicated in a previous RNAi study, in which LOC107398253 ncRNA (decreased 12.3-fold, $p = 0.001$, in the current study) was decreased 593-fold in response to RNAi of *ADC* (*Perkin, Gerken & Oppert, 2017b*). This particular ncRNA was annotated as an U3 snoRNA, predicted to be involved in site-specific cleavage of ribosomal RNA (rRNA) during pre-rRNA processing (*Clery et al., 2007*). More work is needed to understand the relationship of ncRNAs and RNAi in insects.

## CONCLUSIONS

In summary, these data add to the sparse stage-specific studies in *T. castaneum* and other beetles. We demonstrate that by using this approach, we were able to identify a candidate gene, CPG, that may be useful in developing an insect control product based on dsRNA. Because the gene was expressed at low levels overall and moderate levels in the larval gut, oral delivery methods for CPG dsRNA have the potential to control damage by *T. castaneum* larvae.

## ACKNOWLEDGEMENTS

The authors would like to thank technicians Ken Friesen and Tom Morgan for their contribution, injecting larvae and RNA extraction and library preparation, respectively. Mention of trade names or commercial products in this publication is solely for the purpose of providing specific information and does not imply recommendation or endorsement by the US Department of Agriculture. USDA is an equal opportunity provider and employer.

### Funding

The authors received no funding for this work.

### Competing Interests

Brenda Oppert is an Academic Editor for PeerJ.

## Author Contributions

- Lindsey C. Perkin conceived and designed the experiments, performed the experiments, analyzed the data, prepared figures and/or tables, authored or reviewed drafts of the paper, approved the final draft.
- Brenda Oppert conceived and designed the experiments, analyzed the data, contributed reagents/materials/analysis tools, authored or reviewed drafts of the paper, approved the final draft.

## Data Availability

The *T. castameum* life stage sequences used in the first analysis described here are accessible via the NCBI SRA BioProject number PRJNA299695. The CPG RNAi knockdown sequences are accessible via the NCBI SRA BioProject number PRJNA520884.

## Supplemental Information

Supplemental information for this article can be found online at http://dx.doi.org/10.7717/peerj.6946#supplemental-information.

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
