# Peer review of "Gene expression in Tribolium castaneum life stages: Identifying a species-specific target for pest control applications"

_PeerJ, doi:10.7717/peerj.6946_

## Round 0.1 · original submission · Major Revisions

While most reviewers' comments are editorial, some are more substantial, especially those regarding multiple testing and significance/effect analysis. Further, the logic of some steps is not clear, as presented, and should be explained in more detail and with additional motivation.

·

Basic reporting

In general paper is good written and all necessary background is provided. But there are some ambiguities:
1) From manuscript is it unclear what RNA-Seq data were generated for this study and what have been already published. As far as I can understand life stage data were published previously, while RNA-Seqs for CPG knockdown were generated in this study. But there is no information about whether (and where) the latter data made public.
2) Authors should clearly explain what expression threshold they used. "RPKM 2>8 log2" is completely meaningless for me.
3) Authors should clearly explain how they define significant GO terms. "reduced to the most specific" is not enough.
4) From the method section it seems that authors didn't use multiple texting correction when they analysed RNA-Seqs for CPG knockdown. Authors should either explain how they corrected for multiple testing or implement such correction if it was not done yet.
5) Authors are encouraged to illustrate results of differentiall expression analysis using volcano plots or by similar visualizations methods that shows both: significance and effect size.
6) It is unclear why authors think that best targets should have low and stage-specific expression. Naive view suggest the opposite... Authors should clearly explain their motivation to select target in that way
7) Target selection procedure should be clearly explained including all sequentially applied threshold and filters altogether with number of genes passed given threshold. Authors should provide motivation for all thresholds used (for example histograms of parameters used for filtering)
8) On line 412 Authors state that they found not alignment of part of CPG gene in mRNA of other species. But it may be caused by differences in annotation, probably this part of gene is not annotated in other species. Search should be performed against whole genome (probably only syntenic region), not just mRNA.

Experimental design

While I'm not a specialist in experimental biology, I believe that Mock should contain not only dye but also dsRNA with shuffled sequence. In the current design it is not possible to distinguish effect of CPG knockdown from "any double RNA ejection"

Validity of the findings

Taking into account my comments on experimental design and reporting, for now it is unclear whether results are well supported.

Reviewer 2 ·

Basic reporting

The language of this paper is mostly clear and unambiguous, with only a few exceptions, which are noted below. Professional English is used throughout.

The references used are adequately extensive. That said, the authors reference the iBeetle Project, which has aimed to collect phenotypic data from RNAi knockdown of all Tribolium genes. However, the authors did not address if the iBeetle Project had also targeted CPG. I think the authors need to address this prior to publication. Even if the iBeetle Project had tested knockdown of CPG, the results are worth repeating, particularly with the addition of the authors’ RNA-seq data, but what, if any, findings the iBeetle Project got from CPG should also be discussed, lest readers suspect the authors of hiding something. If the iBeetle Project has not examined CPG, this should be state clearly.

The structure, figures, and tables of this article were predominantly professional and well-managed, with a few exceptions: Figures 1 and 3: Is the intensity of the grey color in these figures significant? If so, this should be explained in the legends. This work is self-contained and coherent.

As mentioned above, I found several very minor issues with the language and organization that I think would improve the readability of the paper. I will list them in the order they were found in the paper:

A: Abstract:
Lines 29-30: the list of genes do not appear to offer any contribution to the abstract, especially considering none of these genes is discussed in much greater detail in the paper. The abstract would benefit from leaving them out.
Line 33: the phrase “284-fold” makes this sentence awkward, and can easily be left out.
Line 35: “RNA” should be plural. There were several other instances throughout the paper where plural noun forms were not used consistently. I encourage the authors to double check for appropriate and consistent use of plural/singular.

B: Introduction:
Lines 75-76: Prior to this sentence, the authors mention the importance of using less conserved genes when planning RNAi for pest control. They then describe how results from the iBeetle study, which was carried out in Tribolium, were used as the basis of an RNAi study in D.v.virgifera. What is not clear is if the authors are suggesting the D.v.virgifera study proves their point about genetic conservation resulting in off-targets, or if they are lifting up the iBeetle project as an example of how research in Tribolium can be applicable to other species. This sentence should be rephrased, or the paragraph reorganized to clarify this issue.
Line 87: the word ‘resistance’ would probably be clearer than the word ‘resistant’ when describing genes that help insects escape the effects of pesticides.
Lines 97-99: This sentence is worded awkwardly. Replacing the phrase “reducing the expression of CPG” with something simpler, like “CPG RNAi” could improve the readability. Also, I believe the word “connection” would make more sense to most readers, rather than “connectivity”

C: Materials & Methods:
Line 129: I believe the authors intended “RPKM<8 log2” not “RPKM 2>8 log2”

D: Results:
Line 237 ff: The authors are inconsistent in their use of plurals when referring to life stages. I encourage them to be consistent. “Eggs” and “Adults” seemed particularly prone to loss of the plural form.
Lines 255-262: this paragraph seemed out of place – out of the flow of the narrative the authors had been establishing with their results. I just want to encourage the authors to reexamine the placement of this paragraph, to ensure that it is in the right place. If so, they may benefit from the addition of a transition to help readers follow their line of thought in this matter.
Line 299: the authors need to add a close parenthesis, “)”, after the word “genes”
Line 313: The authors should add “p=” to the beginning of the second parenthetical.

E: Discussion:
Line 391: “male” and “female” should probably be plural.
Line 422: the arrangement of the phrase “non-injected Control and Mock” could confuse readers into believing that both “Control” and “Mock” are “non-injected.” I would recommend that the order be swapped; alternatively, the phrase “non-injected” could be dropped, since “Control” have already been defined as “non-injected” elsewhere.

Experimental design

The authors do a good job of explaining the questions they are trying to address with this research. Their described methods seem thorough, with only a few exceptions describe below. Their experiments were well designed to address their questions in a rigorous way.

Line 162: the authors describe one PCR step as being “anneal at 68C.” I believe they mean “extension” here, not anneal. They also describe their conditions as “standard”, but I believe 72C for extension is more typical. Unfortunately, the authors do not reveal the polymerase they used. Is 68C the required extension temperature for their polymerase, or are they using an unconventional temperature for reasons they then fail to explain? Either way, more detail is needed for this part of the M&M

Line 223: The version of MEGA used is several versions old. While the authors did not get any unusual results in their trees, my experience is that more recent versions are more accurate. While I wouldn’t expect the tree to change significantly, a more recent version could help with the shockingly low bootstrap values seen near the bases of the tree. I encourage the authors to explore this possibility, although I do not think publication of this paper should depend on them doing so.

Validity of the findings

The authors describe their results thoroughly, and do a great job of not over-selling their conclusions. The data and analyses appear to be very sound.

Additional comments

SUMMARY of the PAPER: The authors of this paper report on the use of RNA-seq to identify gene expression changes among the major life stages of the beetle, Tribolium castaneum. They found several gene expression changes that may be of interest to the larger Tribolium research community, particularly for those interested in gene-based methods of pest control. Indeed, as a proof of principle of their methods, the authors identified a fairly unique gene, CPG, with a low, larval expression pattern as a target for RNAi. Reduction of target expression enhanced lethality in injectees. RNA-seq data was then collected from RNAi individuals to examine gene expression changes resulting from knockdown of CPG; these results hint at a mechanism for escape from the lethal effects of CPG knockdown.

---

## Round 0.2 · Minor Revisions

The revievers have listed a number of remaining concerns that should be addressed.

·

Basic reporting

Paper is good written and all necessary background is provided. Unfortunately, I'm still cannot understand notation authors used to denote expression thresholds: "RPKM greater than 2 less than 8 log2". Does it mean that log2(RPKM) should be between 2 and 8?

Experimental design

In revised version authors didn't address two of by concerns:
1) Authors didn't use shuffled RNAi control. In rebuttal they said that use of such controls "caused a lot of problems", that they can (but did not) use "scramblase" control, and that later should be very similar to Mock. I do not understand the difference between shuffled RNAi and "scramblase" as well as I do not understand why first should "cause problems" - to my knowledge it is widely used. I believe that this issue should be reviewed by someone with strong experimental background.
2) Authors didn't prove species-specificity of sequence used as RNAi target, in rebuttal they told that they run some searches (presumably using some web-services) but unsuccessfully because of time out. Just to note, it is just a technical problem - blastn (as well as other tools used) can be run locally. This issue is not so important, since species-specificity of designed RNAi is not a main topic of the paper.

Validity of the findings

With exception of one concern mentioned above, I'm satisfied by experimental design an think that conclusions are supported by shown results

Reviewer 2 ·

Basic reporting

The language of this paper is mostly clear and unambiguous, with only a few exceptions, which are noted below. Professional English is used throughout.

- Lines 72-73: The authors say "secondary primers were designed to avoid off-target effects using OfftargetFinder." Because the authors had already mentioned OfftargetFinder, this sentence was confusing. I believe the authors mean that they designed the secondary primers to amplify the region of minimal off-target effects that they had identified with OfftargetFinder, but this sentence could be edited to be clearer about this.

- In lines 301-302 and 308, the authors appear to list multiple GO terms, but list all the terms within a single pair of quote marks. If these terms are indeed separate, each term should be given its own pair of quote marks.

- Line 341: the authors say "fold-change", but I think they mean "expression"

The references used are adequately extensive. The structure, figures, and tables of this article were predominantly professional and well-managed, with a few exceptions:

- In figures 2 and 4, the authors say "The number in parentheses indicates the number of genes in each category." However, no parenthetical numbers were visible in the actual images.

Experimental design

The authors do a good job of explaining the questions they are trying to address with this research. Their described methods seem thorough, and their experiments were well designed to address their questions in a rigorous way.

Validity of the findings

The authors describe their results thoroughly, and do a great job of not over-selling their conclusions. The data and analyses appear to be very sound.

That said, in lines 342-344, the authors discuss the significant mortality caused by CPG RNAi. They discuss the significance only relative to Control, but also mention the Mock mortality, only to say it was "not significant." Presumably, they are stating that mortality of Mock injected individuals was not significant relative to Control. However, the do not give the mortality rate of Control, which was clearly non-zero, nor do they give the significance between Mock and RNAi injections. Both of these numbers would be helpful to clarify the degree to which CPG RNAi causes death. It might also be helpful, but is not necessary, to discuss the rate of death, since figure 5 seems to suggest ~half of all RNAi deaths came within the first 24 hrs.

---

## Round 0.3 · accepted · Accept

I'm satisfied that most concerns of the reviewers have been addressed in the revised manuscript. There are some valid comments on the overall design of the study, but they do not invalidate the findings.

#